# Viscoelastic Reversibility of Carrageenan Hydrogels under Large Amplitude Oscillatory Shear: Hybrid Carrageenans versus Blends

**DOI:** 10.3390/gels10080524

**Published:** 2024-08-09

**Authors:** Izabel Cristina Freitas Moraes, Loic Hilliou

**Affiliations:** 1Department of Food Engineering, Pirassununga, Faculty of Animal Science and Food Engineering (FZEA), University of São Paulo (USP), São Paulo 13635-900, Brazil; bel@usp.br; 2Institute for Polymers and Composites, University of Minho, 5800-048 Guimarães, Portugal

**Keywords:** hybrid carrageenan, LAOS, strain hardening, mixed carrageenans

## Abstract

The viscoelastic response of carrageenan hydrogels to large amplitude oscillatory shear (LAOS) has not received much attention in the literature in spite of its relevance in industrial application. A set of hybrid carrageenans with differing chemical compositions are gelled in the presence of KCl or NaCl, and their nonlinear viscoelastic responses are systematically compared with mixtures of kappa- and iota-carrageenans of equivalent kappa-carrageenan contents. Two categories of LAOS response are identified: strain softening and strain hardening gels. Strain softening gels show LAOS non-reversibility: when entering the nonlinear viscoelastic regime, the shear storage modulus *G′* decreases with increasing strain, and never recovers its linear value *G*_0_ after successive LAOS sweeps. In contrast to this, strain hardening carrageenan gels show a certain amount of LAOS reversibility: when entering the nonlinear regime, *G′* increases with strain and shows a maximum at strain *γ_H_*. For strains applied below *γ_H_*, *G*_0_ shows good reversibility and the strain hardening behavior is maintained. For strains larger than *γ_H_*, *G*_0_ decreases significantly indicating an irreversible structural change in the elastic network. Strain hardening and elastic recovery after LAOS prevail for hybrid carrageenan and iota-carrageenan gels, but are only achieved when blends are gelled in NaCl, suggesting a phase separated structure with a certain degree of co-aggregated interface for mixed gels.

## 1. Introduction

The food industry makes great use of carrageenan for texturizing formulations, as these natural polymers form viscous solutions or thermo-reversible gels in the presence of water and salts [1]. There is increased interest in these sulfated polysaccharides as films for active food packaging applications [2], and in the pharmaceutical sector [3], where they have recently shown promising solid-state properties for use as excipient in tablet formulations [4].

Carrageenans are present in the cell walls of marine seaweeds, where they confer mechanical resistance to stresses such as waves and desiccation [5]. These biopolymers show a complex chemical structure, which is better described as a multi-block copolymer made of statistically distributed sequences (blocks) of alternating 3-linked β-D-galactopyranose (***G***-units) and 4-linked α-D-galactopyranose (***D***-units) or 4-linked 3,6-anhydrogalactose (***DA***-units) [6]. The position of sulfate groups on the ***G***, ***D***, or ***DA*** units defines the carrageenan type. Among this polysaccharide family, the following gelling carrageenans can be highlighted: kappa-carrageenan, essentially made of sequences of ***G4S-DA*** (here labelled ***κ***); iota-carrageenan, which contains at least 90 mol.% of ***G4S-DA2S*** (here labelled ***ι***); and hybrid carrageenans, which have a more heterogeneous structure ranging from the kappa2-carrageenan (or weak kappa as defined by the industry) to a very heterogeneous structure where sequences of ***κ*** and ***ι*** coexist in the polymer chain with disaccharide units such as nu-carrageenan (***G4S-D2S,6S***, ***ν***) or mu-carrageenan (***G4S-D6S***, ***μ***) [7] (see Figure 1a). 

The gel mechanism in carrageenans is still the subject of a vivid debate, but there is a consensual description for a volume phase transition occurring during the cooling of a hot carrageenan solution in the presence of salt and resulting from the self-assembly of helical conformers into a three-dimensional network [9]. However, the relationships between the network structure and its elasticity are still a subject of fundamental research, see, e.g., [10] and the references therein.

In spite of its relevance for the industry, which processes carrageenan gels under large deformation, their nonlinear rheological behavior has not received much attention in the literature [7,10,11]. Shear softening, where the elastic component *G′* of the shear complex modulus decreases as the shear strain amplitude increases, has been reported for commercial kappa- and iota-carrageenans, as well as their blends [12,13,14]. In contrast to this, other studies on hybrid carrageenans report a strain hardening, that is an increase in *G′* with the increased strain amplitude, which has not received further attention [15] or has been rationalized by colloidal gel theories to assign this nonlinear elastic behavior to the fractal structure of the network [16]. Similarly, efforts to relate the carrageenan gel structure studied by neutron scattering to the large deformation behavior were recently attempted with commercial products [17]. The strain softening of a kappa-carrageenan gel under large amplitude oscillatory shear (LAOS) was confirmed and assigned to the deformation of a frozen network of large rigid aggregates, whereas the strain hardening of an iota-carrageenan gel was attributed to the deformation of a network made of smaller and loose aggregates. In addition, the authors reported the response of both types of gels to two repeated strain sweeps performed in the nonlinear regime of deformation. Such preliminary results suggested the microstructural damage of gels mirrored in the decrease in their viscoelastic properties.

The response of carrageenan gels to repeated strain sweeps, where large deformations corresponding to the strain hardening or strain softening regimes are repeatedly applied, has virtually not been reported in the carrageenan literature. Thus, the main objective of the present work is to report on the LAOS behavior of hydrogels made from the four families of carrageenan systems mostly used in applications: iota-carrageenan gels (I), kappa-carrageenan gels (K), their mixture (K+I), as well as hybrid carrageenan gels (KIMN). The latter are hot water extracted from commercial seaweeds used to produce a range of sulfated polysaccharides: from nearly pure iota- and kappa-carrageenans to kappa2-carrageenans. Because the LAOS behavior of various carrageenan gels is the focus of the present study, the type of salt and ionic strength needed to form a gel with enough elasticity for rotational rheometry for all types of studied carrageenans are to be established. Earlier studies on the phase diagrams of KIMN and K+I blends showed that gels could be formed with a carrageenan concentration of 1 wt.% and 1 M NaCl or 0.1 M KCl [18,19]. Thus, such gelling conditions were used to report on the LAOS response and the LAOS reversibility of the four types of carrageenan gels. In addition, a systematic comparison between the LAOS response of the hybrid carrageenan gels and the LAOS response of the K+I gels of near-similar carrageenan compositions is carried out. This is to shed some light on the structure of the blend, as this topic is receiving recent increased interest (see, e.g., [20] and references therein), in contrast to their LAOS behavior.

## 2. Results and Discussion

### 2.1. Chemical-Physical Characteristics of Hybrid Carrageenans

Figure 1 presents the proton NMR spectra of extracted hybrid carrageenans, whereas Table 1 summarizes their polymer characteristics. Hybrid carrageenans were hot extracted from commercial seaweeds in deionized water, see full experimental detail in Section 4.2. For NMR measurements, solutions of extracts in D_2_O were prepared. To limit line broadening in NMR spectra, these solutions were ultrasonicated for a few hours before filling the NMR tubes and acquiring NMR spectra at 70 °C. Full experimental detail is given in Section 4.2. All extracts are hybrid carrageenans as the peaks assigned to anomeric protons of the kappa-carrageenan family are resolved.

In particular, the KIMN1 sample belongs to the iota/nu-hybrid-carrageenan family, as the amount of ***κ*** and ***μ*** disaccharide units showing up at 5.1 ppm and 5.25 ppm, respectively [8], is close to the limit of sensitivity of the spectrometer. The comparison of KIMN1 with a commercial iota-carrageenan sample (I) allows assessing the effect of few ***ν*** disaccharide units in the polymer chain on the gelling behavior of an iota-carrageenan. A single study reported that after a gel elasticity improvement due to the incorporation of up to 3 mol.% of ***ν*** disaccharide units, the elasticity of the hydrogel drops, since more ***ν*** units act as defects, impeding the helical association of iota-carrageenan chains [21]. The remaining samples show a more complex hybrid character. Samples KIMN2 and KIMN3 can be compared with a mixture of 50 mol.% of kappa-carrageenan with 50 mol.% of iota-carrageenan, especially if one assumes that the kappa-carrageenan chains in the mixture are responsible for the gel elastic behavior, and that iota-carrageenan chains do not have a significant elastic contribution in the mixture under the salt conditions used here [20]. KIMN4 is essentially made of sequences of ***κ*** disaccharide units. When compared with sample K, KIMN4 is used here to question the effects of few sequences of ***ι*** disaccharide units and of disaccharide units of ***ν*** and ***μ*** types on the viscoelastic behavior of the corresponding gels. More interesting is the absence of starch (no peaks at 5.36 ppm and 4.98 ppm [8]) or pyruvic acid in the selected samples, which could interfere with the gelling mechanism of the carrageenan chains.

The molecular weight distributions of extracted hybrid carrageenans are overall similar, if one takes into account the polydispersity reported in Table 1 which is defined here by the ratio of the weight averaged molecular mass *M_w_* over the number averaged molecular mass *M_n_*. All extracted samples are essentially equivalent to the size of the order of 10^6^ g/mol. This is expected for these polysaccharides [11] and for the type of extraction adopted here to keep the biological precursors ***ν*** and ***μ*** in the copolymer chain [22].

The commercial carrageenans (samples I and K) are essentially homopolymers with macromolecular masses of the same order as the KIMN samples. All carrageenans in Table 1 contain a certain amount of salt either originating from the dried seaweeds which were used as received [23], or from the alkali treatment used to produce the commercial K and I samples (see Section 4).

### 2.2. LAOS Behavior: Strain Hardening versus Strain Softening

The responses of gels made in KCl to a sweep in the amplitude of the oscillatory shear strain are presented in Figure 2 and Figure 3 for the three types of carrageenan studied here.

At low shear strains, the linear domain of viscoelasticity (where the moduli are not dependent on the shear strain) allows for the measurement of the (linear) shear elastic modulus *G*_0_ of the gels. The linear domain (labelled region 1) terminates at a critical strain *γ_C_*, which is estimated by the strain value where the shear storage modulus *G′* deviates by 5% from *G*_0_. The measured *γ_C_* for gels formed in 0.1 M KCl are listed in Table 2. Beyond *γ_C_*, the shear storage modulus *G′* increases sharply with the strain for all KIMN samples and sample I (see Figure 2). This indicates that the elasticity of the network increases with strain. A strain hardening behavior is often described for fibrillary biological networks [24,25,26,27], and stems from the transition between the entropic and enthalpic contributions to the stretching of filaments pre-tensioned in a network. This is overall in harmony with earlier rheological results documented with a hybrid carrageenan gel with a chemical structure incorporating ***ν*** and ***μ*** disaccharide units [16] and with a Kappa-2 gel [6], both under similar salt conditions and polysaccharide concentrations. The strain hardening of iota-carrageenan gels has been reported in few rheological studies where different salt and gelling conditions were used [10,17]. In all these reports, no control of the sample thickness was performed, in contrast to the present study where the normal force is maintained at 0 N during the gel setting. This is to accommodate the volume change during the liquid-to-gel transition [28], and to compensate for the thermal expansion of the shearing plates during the cooling that sets the gel in the rheometer.

The strain hardening illustrated in Figure 2 is overall the same for the KIMN3 and I samples, as *G′* passes through a maximum located at a specific strain *γ_H_*, which signals the end of region 2, before a monotonic decrease at larger strain. Note here that the strain hardening depicted in Figure 2 was labelled elsewhere as a strong strain overshoot [29] to qualify the LAOS response of various materials without any relation to the underlying hardening structure. However, a small qualitative difference is evident in the strain dependence of the loss modulus *G″*. For the hybrid carrageenan sample, *G″* exhibits a maximum at strain *γ_M_* (which defines the upper limit of region 3) before the crossover between both moduli, which occurs at a strain *γ_F_* (which defines the end of region 4). In contrast to this, no region 3 is identified for sample I: the strain where *G″* is peaking up coincides with *γ_F_*. The latter signals the strain-induced fluidization of the gel, since beyond this critical strain the material’s loss modulus *G″* becomes larger than the elastic modulus *G′*. Thus, above *γ_F_*, in region 5, the material is essentially viscous, in contrast to the solid character of hydrogels. The shear stress *σ_F_* transmitted by the sheared sample at strain *γ_F_* is reported in Table 2, together with all other rheological parameters which characterize the LAOS behavior of all studied gels.

The LAOS behavior of the kappa-carrageenan gel portrayed in Figure 3 is different from the ones displayed in Figure 2. Such behavior was labelled as a weak strain overshoot elsewhere [29]. Beyond *γ_C_*, *G′* decreases with the shear strain, whereas *G″* passes through the maximum. This is reminiscent of the shear softening behavior previously reported for other commercial kappa-carrageenan gelled at various salt and polysaccharide concentrations [12,13,14,17].

The mixture of kappa- and iota-carrageenan exhibits a qualitatively similar shear softening behavior (see Figure 4), as expected from a previous study carried in blends gelled in 0.03 M KCl [17]. However, the rheological parameters listed in Table 2 show large quantitative differences when compared with the parameters measured with the kappa-carrageenan gel. Indeed, the blend is more resistant to the shear strain than the kappa-carrageenan, since *γ_F_* is significantly larger. This enhancement of the gel’s strain resistance is in harmony with earlier studies [30].

The quantitative analysis of the LAOS behavior presented in Table 2 also offers a discussion of the effects of the gel structure on the rheological properties. Blending gives a very different rheology than opting for a hybrid carrageenan to formulate the hydrogel. The gel from the mixture is significantly stiffer than hybrid carrageenan gels (compared to *G*_0_), which results from the kappa-carrageenan network in the phase separated [17,30,31,32], interpenetrated [33], or co-aggregated [20,34] gel structure. However, the copolymers show strain hardening and less resistance to shear strain (compare *γ_F_*). Such differences support a phase separated structure in kappa- and iota-carrageenan blends (which, in our understanding, also includes the interpenetration of two separated networks) as blocks of ***κ*** disaccharide units cannot separate from blocks of ***ι*** disaccharide units in the KIMN samples. The elasticity *G*_0_ of the blend can be compared with the computation from the *G*_0_ of the pure components following the isostrain or the isostress approaches (see [30] and the detailed discussion therein). The isostress approach, which shows conceptual similarities to a linear viscoelastic Maxwell model for the fluids, hypothesizes a mechanical link between the two networks. Such coupling inherently leads to the mechanical preponderance of the less elastic network: *G*_0_ of K+I would be of the order of 116 Pa found for sample I. Alternatively, the isostrain approach assumes no interaction between the two independent networks, and thus shares similarities with a linear Kelvin–Voight model for viscoelastic solids. In this case, the kappa-carrageenan network rules the elasticity of the blend since it is the sum of the two networks formed at 0.5 wt.%. Assuming a power law with an exponent 1.5 for the concentration dependence of *G*_0_ for both iota- and kappa-carrageenan gels (an exponent found for rigid polymer networks where the elasticity comes from the enthalpic contributions of frozen stranded crosslinks [35]), the sum of the two networks formed at 0.5 wt.% gives a total *G*_0_~7420 Pa, which is compatible with the values of *G*_0_ reported in Table 2.

A few non-gelling ***ν*** disaccharide units in an iota-carrageenan chain are responsible for an elastic weakening of the network (compare samples KIMN1 and I in Table 2), whereas the strain hardening is enhanced (compare the difference | *γ_C_* − *γ_H_* | which quantifies the strain range where the gel hardens) as well as the overall strain resistance before fluidization (see *γ_F_*). Molecular theories for the strain hardening of networks of filaments suggest that filaments in KIMN1 are less pre-compressed upon crosslinking during gel setting than are iota-carrageenan filaments, and thus shows smaller *G*_0_ and larger stiffening [27]. The question, however, remains whether such difference is due to the ***ν*** disaccharide units acting as defects in the iota-carrageenan network or to the fact that 4 mol.% ***κ*** disaccharide units is not a sufficient fraction to reinforce the iota-carrageenan network through co-aggregation as in sample I with 8 mol.%.

### 2.3. LAOS Reversibility

The mechanical reversibility of the gels submitted to successive LAOS sweeps initiated in the linear regime is displayed in Figure 4 for two illustrative carrageenan networks prepared in 0.1 M KCl.

The strain hardening iota-carrageenan gel exhibits nearly full viscoelastic reversibility in the whole range of tested strains, whereas the strain softening K+I blend shows a quite opposite trend. As expected for both samples, right after applying a first strain sweep in the linear regime (sweep 1, with strain amplitude *γ* < *γ_C_*, not plotted in the figure as it cannot be distinguished from the remaining sweeps), the second strain sweep (sweep 2) delivers a fully reproducible linear viscoelastic response. The reproducibility in sweep 2 includes the critical strain *γ_C_*, since the values of *γ_C_* inferred from the data in Figure 4 are identical to the values listed in Table 2 and highlighted in Figure 4. Thus, sweep 2 also shows the full reproducibility of the data measured with two different samples gelled in the rheometer. Right after sweep 2, a third sweep (sweep 3) is applied with strain ranging from region 1 (linear regime) to the onset of region 4, where all samples are strain softening. In sweep 3, the iota-carrageenan gel shows full viscoelastic reversibility (in both *G′* and *G″*), encompassing the linear viscoelastic modulus *G*_0_, the early strain hardening regime and the value for *γ_H_*. In contrast, the gel made from the blend does not recover its *G*_0_, shows a wider linear regime but G″ presents a maximum at nearly 15%, which is smaller than *γ_M_*. Likewise, the following strain sweeps (sweeps 4 and 5 with strains ranging from 0.1% to regions 4 and 5, respectively) indicate a partial breakdown of the K+I gel structure as the gel elasticity *G*_0_ drops by one order of magnitude, whereas the strain softening and the fluidization occur at smaller strains than those reported in Table 2. The iota-carrageenan gel only shows elastic irreversibility in the nonlinear regime as *G*_0_ is maintained for all sweeps. Structural changes in the gel network during sweeps 4 and 5 are only mirrored in the values of the strains where *G′* is maximum during hardening, and in the drop of *G′* beyond this critical strain. Remarkably, the loss modulus *G″* shows a full viscous reversibility as its maximum and the fluidization are almost reproduced at *γ_M_*~*γ_F_*.

The effect of the hybrid carrageenan chemical structure on the LAOS reversibility is presented in Figure 5 for selected gels. To better highlight the elastic recovery or non-reversibility of the gels, the strain dependence of the scaled storage moduli *G′/G*_0_ are presented, as the strain dependences of *G″* during each sweep essentially conveys the same information. In the scaling, *G*_0_ is taken from the linear regime of the first sweep.

Figure 5a indicates that the presence of up to 17.9 mol.% of defects on the iota-carrageenan chain does not jeopardize the excellent mechanical reversibility found in Figure 4 for the I sample. The main difference between samples I and KIMN1 is the quantitative drop of 10% in *G*_0_ for the KIMN1 gel after applying a strain *γ* > *γ_H_*, which contrasts the fully reversible elasticity of the iota-carrageenan gel. The KIMN2 sample also maintains the strain hardening behavior, and *G*_0_ drops only by 47% after fluidizing the gel by applying a strain *γ*~*γ_F_* during sweep 4. The elastic nature of the network in the hybrid carrageenan gels is thus essentially maintained, but the strain hardening assigned to the iota carrageenan blocks in the copolymer has vanished during the flow-induced restructuring of sweep 4. The structural resistance to strain is virtually lost for KIMN4, which shows a LAOS non-reversibility equivalent to a kappa-carrageenan gel. This is illustrated in Figure 6, where the drop in *G*_0_ is as much as 90% after sweeping the strain up to *γ*~*γ_F_* for both gels. However, Figure 6 highlights the main qualitative difference between these two gels: the strain hardening behavior of KIMN4, which shows up on a limited shear strain range.

Interestingly, sample KIMN4 shows a much smaller strain hardening than the one measured under same salt and carrageenan concentrations with a hybrid carrageenan made of 70 mol.% ***κ*** and 30 mol.% ***ι***. [6]. Together with the results presented in Figure 5 and Figure 6, this suggests that at least 27.6 mol.% ***ι*** are needed in the hybrid carrageenan to endow the gel with more than 50% elastic recovery after strain-induced fluidization, whereas only 16.2 mol.% ***ι*** are sufficient to impart strain hardening. Adding up to 50 wt.% iota-carrageenan in a blend with kappa-carrageenan will not confer such mechanical properties to the gel, since the K+I sample is irreversibly strain softening (see Figure 4). In future studies, it would be interesting to find out the minimum iota-carrageenan amount in the blend needed to endow the gel with iota-carrageenan-like LAOS behavior.

### 2.4. Effect of Salt

Kappa-carrageenan is known to form stronger gels in KCl than in NaCl, whereas the gel properties of pure iota-carrageenan show no cation sensitivity [32,36]. It is thus expected that K+I blends, hybrid carrageenans, and sample I containing 8 mol.% will show specific LAOS behavior and strain reversibility when gelled in the presence of NaCl. The iota-carrageenan gel shows a less pronounced strain hardening than in KCl, as can be seen from the comparison of the data collected in Table 3 for NaCl and Table 2 for KCl. *|γ_C_ − γ_H_|* is small or within the precision of the steps between consecutive strains, and the increase in modulus (the hardening) is of the order of 5%. In addition, the application of a sweep within region 4 is challenging as *γ_M_*~*γ_F_* (see Table 3). Thus, no elastic recovery after LAOS performed up to *γ_F_* can be inferred from the data, whereas full reversibility in *G*_0_ is found after sweeping the strain up to *γ_H_*, much like in KCl. Nevertheless, a small effect of the type of cation on the linear (*G*_0_ slightly smaller) and LAOS behavior (less hardening) is found, which can be related with the presence of 8 mol.% of ***κ*** disaccharide units in the iota-carrageenan chain if one extends the linear viscoelastic results found in [32] to the LAOS regime.

Alternatively, the role of ***κ*** disaccharide units in the iota-carrageenan chain on the cation specificity can be questioned in light of the results found with sample KIMN1 (see Figure 7a). Like for sample I, a cation-induced weakening of the gel strain hardening is evidenced: *γ_C_* cannot be distinguished from *γ_H_* (see Table 3).

Overall, the comparison of results from samples KIMN1 and I in both salt conditions indicates that defects in the iota-carrageenan chains, introduced either at the disaccharide level (kink due to ***ν*** disaccharide units in sample KIMN1), or at the helical level (***κ*** disaccharide units which form helices in the network [37]), have an equivalent impact on the partial loss of the strain hardening of the iota-carrageenan network induced with either NaCl of KCl. Because of the clear difference between *γ_M_* and *γ_F_*, the LAOS reversibility can be studied for KIMN1. An elastic reversibility of 80% for *G*_0_ could be measured right after sweeping the strain up to nearly *γ_F_*, whereas the strain hardening regime vanished right after the sweep entered region 3. This contrasts with the strain hardening reversibility of this same sample gelled in the presence of KCl (see Figure 5a).

For the kappa-carrageenan sample, the effect of the cation is mirrored in the smaller elasticity of the gel and in a smoother strain softening behavior under LAOS. *G*_0_ is more than 20 times smaller in NaCl than in KCl, compare Table 2 and Table 3. Accordingly, this weaker gel is slightly more resistant to shearing, as both the maximum *G″* and shear-induced fluidization are shifted to larger strains, and *G*_0_ recovers 17% (instead of 11% with KCl) of its initial value after sweeping the strain in the softening region 3. The LAOS reversibility of KIMN4 gelled in NaCl is presented in Figure 7b highlighting the viscoelastic role of up to 28 mol.% more sulfated monomers on the network built by kappa-carrageenan chains. NaCl actually enhances the nonlinear viscoelastic effects underpinned with gels formed in KCl: the strain hardening brought by these monomers is more important as |*γ_C_* − *γ_H_*| is much larger than in KCl. On the other hand, the effect on *G*_0_ is weaker, and the LAOS reversibility remains nearly unchanged as parameter *R* is almost the same. Thus, the data in Figure 7b lead to the same conclusion reached earlier when using KCl: 16.2 mol.% ***ι*** disaccharide units on the kappa-carrageenan chain brings essentially nothing more to the gel than strain hardening at the cost of reduced elasticity.

As expected from earlier studies that systematically compared the phase diagrams of blends and hybrid carrageenans with equivalent carrageenan compositions [18,19], no gel was produced with samples KIMN2 and KIMN3 under the tested NaCl gelling conditions. This is due to the presence of the significant amount of ***ν*** and ***μ*** disaccharide units, as shown elsewhere [10,16]. The K+I sample formed a gel with a measurable elastic modulus *G*_0_. More interesting is that the blend now shows both the strain hardening and LAOS reversibility which can be attributed to the iota-carrageenan component (see Figure 7c). The mismatch between the hybrid samples and the blend is again in favor of a phase separated structure in the blend. The iota-carrageenan network now drives the LAOS behavior and the LAOS reversibility (81% of the value of *G*_0_ is recovered after applying a strain sweep well into region 3), whereas the kappa-carrageenan network brings larger elastic contributions to *G*_0_ (see Table 3). An isostrain computation similar to the one performed for KCl above overestimates *G*_0_ (a value of 268 Pa is computed). However, taking now a quadratic increase in *G*_0_ with the carrageenan concentration (as found in many reports, see, e.g., [10]), a total elasticity of 190 Pa is computed for the blend which is a better match to the *G*_0_ listed in Table 3. However, within the picture of two phase separated networks, to reach an elastic modulus of 255 Pa at *γ_H_* implies a nearly tenfold strain hardening of the iota-carrageenan network from 30 Pa (value of *G*_0_ computed for 0.5 wt.% iota-carrageenan with a quadratic concentration dependence). Though such important hardening has been found in several biological networks [25], the largest stiffening reported to date for a carrageenan gel is of the order of twice *G*_0_ [16]. Therefore, the iota-carrageenan network alone cannot contribute to the strain hardening displayed in Figure 7c while the kappa-carrageenan network is strain softening as suggested by sweep 5 (see small drop in *G′*/*G*_0_ before a hardening). We are left with a mechanical coupling between the two networks, perhaps of the kind of helical co-aggregation claimed elsewhere [18,32], to explain the curves in Figure 7c. Indeed, for phase separating polymers, most of the rheology is related to the elastic contributions of the interface between the two components (see for instance [38] for a review of the rheology of immiscible polymer blends). The coupling between the iota- and the kappa-carrageenan networks could share similarities with the strong connections between the mechanically weaker fractal clusters making up colloidal gels that show strain hardening behavior [39]. Such structural and elastic similarities with colloidal gels were suggested earlier for a series of hybrid carrageenan gels [16].

## 3. Conclusions

The large deformation behavior of carrageenan hydrogels submitted to successive LAOS sweeps indicates that hybrid carrageenans and iota-carrageenan share common mechanical features when gelled in the presence of KCl or NaCl. These gels show strain hardening behaviors under large shear deformation, a mechanical characteristic inherent to filamentous networks. The strain hardening is reversible, i.e., the gel linear elasticity can be recovered and the strain hardening may be reproduced, unless strains larger than *γ_H_* corresponding to the maximum in *G′* are applied. The LAOS reversibility is superior for hybrid carrageenans containing more iota-carrageenan disaccharide units on their chain because iota-carrageenan gels are always strain hardening and show excellent strain resistance. In contrast, kappa-carrageenan gels are strain softening and show no LAOS reversibility: when entering the nonlinear viscoelastic regime, *G′* drops irreversibly. However, introducing only 24 mol.% of non-kappa-carrageenan disaccharide units in the block copolymer brings strain hardening with mechanical resistance. Mechanical improvement is also achieved by mixing 50% iota-carrageenan and 50% kappa-carrageenan, but only in NaCl, since the blend is sensitive to salt, although much less than the corresponding hybrid carrageenan, which simply does not gel in NaCl for the tested conditions. The systematic comparison between the LAOS responses of the blends and those from the hybrid carrageenan suggests a phase separation between the two components of the blend, which provides elasticity (from the kappa-carrageenan network), as well as the LAOS reversibility (from the iota-carrageenan network). The phase separated microstructure entails an interface that provides strong links between the two networks. These links, and thus the interface, are necessary for the strain hardening behavior. The nanostructure of the interface will be the subject of future studies. Gels from different blends will be compared to hybrid carrageenan equivalent gels, that is, without ***ν*** or ***μ*** disaccharide units on the chain. This study should rely on the physical interpretation of LAOS characterization [40] and on the rationalization of the strain hardening by molecular theories for filamentous networks.

## 4. Materials and Methods

### 4.1. Materials

Four different KIMN samples were extracted from commercial dried red algae donated by Cargill and used as received. Commercial samples of K (lot 0001432063; product code number 100974290) and I (lot 110M1861V; product code number 1000976834) were purchased from Sigma-Aldrich Química SL (Sintra, Portugal) and used without further purification. Sodium chloride and potassium chloride, both of analytical grade, were purchased from Rotoquimica (Moreira-Maia, Portugal).

### 4.2. Carrageenan Extraction and Characterization

Dried seaweed (1.5 g) was soaked in 100 mL of distilled water at 80 °C for 2 h. Following this, the algal suspension was homogenized in a blender and the extraction process was continued at 90 °C for an additional hour. After this, the centrifugation of the viscous suspensions at 8000 rpm for 10 min was carried out to recover the liquid phase, which was subsequently dried in an oven at 50 °C overnight. Extractions were performed at least in triplicate to produce the necessary amount of carrageenan for following characterizations.

The carrageenan compositions of KIMN extracts and commercial samples were determined by ^1^H nuclear magnetic resonance (NMR) spectroscopy at 70 °C using a Bruker Avance III spectrometer (Billerica, MA, USA) at 400 MHz. Briefly, carrageenan solutions (0.5 wt.% in D_2_O) were ultra sonicated with a DCG-300H bath (MCR Ltd., Holon, Israel) during several hours to lower the viscosity before NMR spectroscopy. The molar fractions (mol.%) of the carrageenan disaccharide repeating units (***κ***, ***ι***, ***μ***, and ***ν***) are calculated as the integrated intensity of the corresponding ^1^H NMR peak (5.10 ppm, 5.28 ppm, 5.5 ppm, and 5.25 ppm, respectively [8]) over the sum of integrated intensities of all assigned carrageenan anomeric protons.

The molecular mass distribution was obtained by size exclusion chromatography (Waters 600 apparatus, with a Waters 2410 differential refractive index detector, Waters Portugal, Lisboa, Portugal) equipped with a PolySep-GFC-P Linear column (Phenomenex, Alcobendas, Spain) and calibrated with pullulan (Shodex, Munich, Germany) ranging from 6300 to 642,000 g/mol. Duplicated measurements were performed with 0.1 wt.% carrageenan solutions in 0.1 M NaCl at 40 °C.

The salt compositions of KIMN samples and commercial carrageenans were estimated from the diffraction spectra of backscattered electron from carrageenan film samples. Hot 1 wt.% carrageenan solutions in deionised water were casted in plastic molds and dried overnight in a ventilated oven to obtain films. The films were then analyzed with a NanoSEM–Nova 200 microscope (FEI, Hilsboro, OR, USA) coupled to a EDAX Pegasus X4M energy dissipative X-ray spectrometer (Ametek, Leicester, UK). All samples contained between 8.7 and 1.4 wt.% K; 4 and 0.7 wt.% Ca; 1.6 and 0.2 wt.% Mg; and 6.8 and 3.7 wt.% Na, which all play no role in the ionic strength of the carrageenan solutions in 0.1 M NaCl or 1 M KCl.

### 4.3. Rheometry

KIMN, kappa-, iota-carrageenan, and their blend (50:50 in weight) were dissolved (1 wt.%) in 0.1 M KCl or 1 M NaCl at 85 °C for 1 h under magnetic stirring. The solutions were left to stand at room temperature overnight. Both ionic strengths and polysaccharide concentrations were selected from previous phase diagrams obtained in KCl and in NaCl for other commercial blends and hybrid carrageenans [35,36] to ensure the formation of gels with elastic properties suited for rotational rheometry.

Hot solutions were gelled from 85 to 25 °C in the parallel plate geometry (40 mm diameter) of a stress-controlled rheometer (MCR 302, Anton Paar, Graz, Austria) and dodecane was used at the rim of the sample to prevent water loss. During the cooling, the sample gap was monitored by the normal force set to 0 N and thus varied between 230 and 280 microns. The normal force-controlled mode allows the measurement of artefact free moduli *G′* and *G″* [26]. After the cooling, small amplitude oscillatory shear (SAOS) measurements were performed to first assess the equilibrium of the gel structure, with a time sweep performed at 25 °C, 1 Hz, and with 0.01% strain amplitude. The dynamic moduli remained time-independent within 1 h, approximately. Subsequently, a logarithmic strain sweep (20 points measured per decade) was conducted in the range of 0.01 to 1000%, at 25 °C and 1 Hz. Analyzing the strain dependence of *G′* and *G″* from this test allowed for the identification of four critical strains, *γ_C_*, *γ_H_*, *γ_M_*, and *γ_F_*, which were used to define the limits of strain sweeps performed with a second sample, gelled and equilibrated as described above. *γ_C_* is the strain signaling the end of the viscoelastic regime, *γ_H_* is the strain where *G′* shows a maximum in the strain hardening regime, *γ_M_* is the strain where *G″* shows a maximum (either during hardening or softening), and *γ_F_* is the strain where *G′* = *G″*, with the corresponding stress *σ_F_*. *R* is the elastic recovery of the gel defined by the ratio *G_5_/G*_0_, where *G_5_* is the linear storage modulus measured right after sweeping the strain up to *γ_F_*. The successive strain sweeps (20 points per decade, logarithmic sweeps performed at 1 Hz) were conducted in the following order, with no time lag between the successive sweeps. First, sweep 1 applied in the linear viscoelastic regime, from 0.01% to a strain *γ* < *γ_C_*. Second, sweep 2 with the strain varied from 0.01% up to *γ_C_*. Third, sweep 3 performed from 0.01% to *γ_H_*. Fourth, sweep 4 where the strain was ramped from 0.01% to *γ_F_*, and a fifth strain sweep (sweep 5) from 0.01% to 1000%.

## Figures and Tables

**Figure 1 gels-10-00524-f001:**
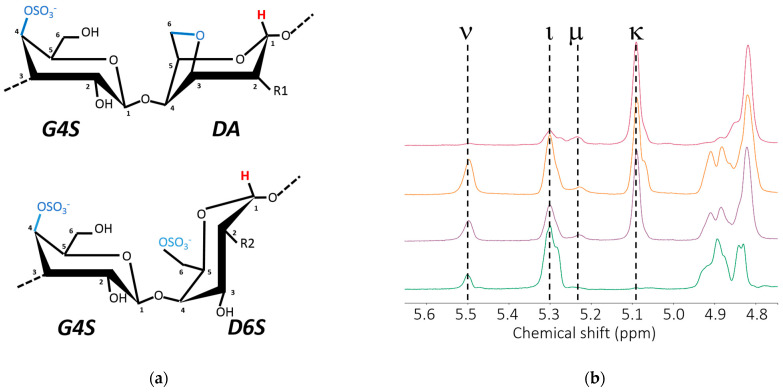
(**a**) Chemical structure of disaccharide units from the kappa-carrageenan family: kappa-carrageenan (R1: OH), iota-carrageenan (R1: OSO_3_^−^), mu-carrageenan (R2: OH), and nu-carrageenan (R2: OSO_3_^−^). (**b**) ^1^H NMR spectra of extracted hybrid carrageenans. From bottom to top KIMN1, KIMN2, KIMN3, and KIMN4. Vertical dotted lines indicate the peaks of the assigned disaccharide units showing up at chemical shifts referenced in [8].

**Figure 2 gels-10-00524-f002:**
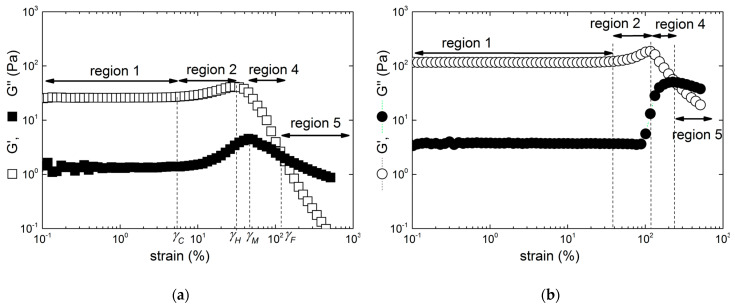
Strain hardening behavior of gels formed in 0.1 M KCl with 1 wt.% carrageenan: (**a**) measured with sample KIMN3; (**b**) measured with sample I, with the maximum in *G″* coinciding with the crossover point between the two moduli, which indicates the absence of a region 3.

**Figure 3 gels-10-00524-f003:**
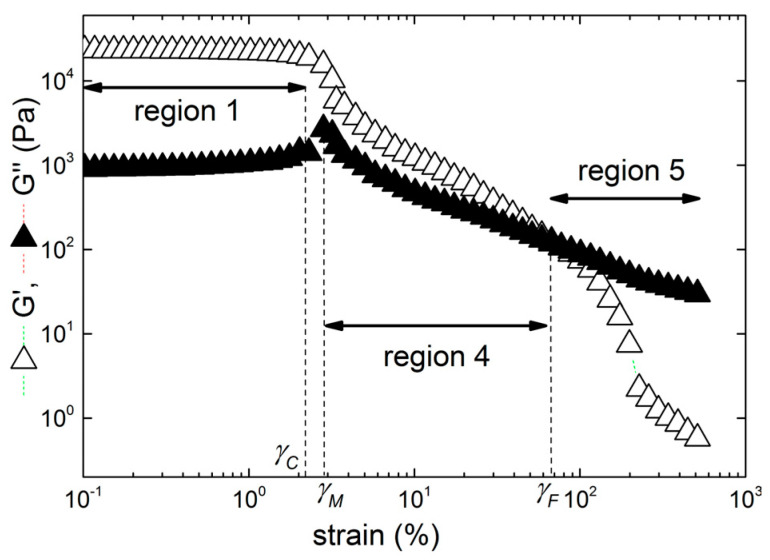
Strain softening behavior of a commercial kappa-carrageenan gel formed in 0.1 M KCl with 1 wt.% polysaccharide.

**Figure 4 gels-10-00524-f004:**
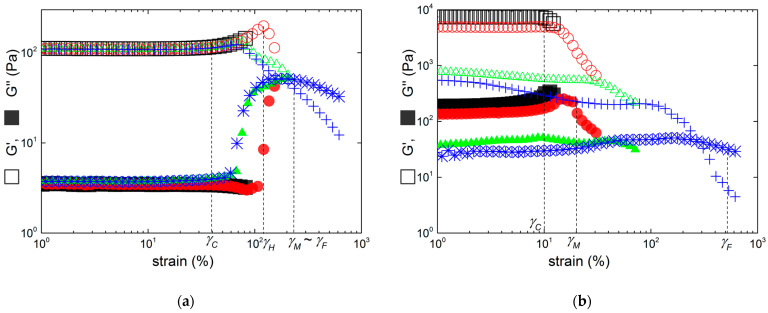
Successive LAOS sweeps performed on carrageen gels in 0.1 M KCl. Each symbol and color indicate a strain sweep and data are presented from a strain of 1% upward to keep visual clarity: (**a**) sample I, sweep 2 from 0.1% to *γ* > *γ_C_* (squares), sweep 3 from 0.1% to *γ* > *γ_H_* (circles), sweep 4 with 0.1% < *γ* < *γ_M_* (triangles), and sweep 5 from 0.1% to *γ* > *γ_M_*~*γ_F_* (crosses); (**b**) sample K+I, sweep 2 from 0.1% to *γ* > *γ_C_* (squares), sweep 3 from 0.1% to *γ* > *γ_M_* (circles), sweep 4 with 0.1% < *γ* < *γ_F_* (triangles), and sweep 5 from 0.1% to *γ* > *γ_F_* (crosses). All critical strains are those measured in a separate experiment and reported in Table 2.

**Figure 5 gels-10-00524-f005:**
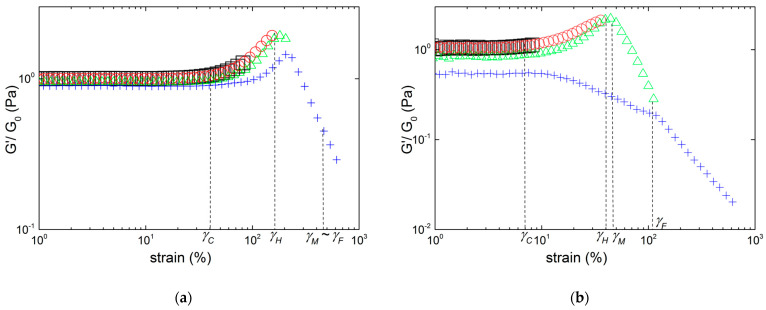
Successive LAOS sweeps performed on hybrid-carrageen gels in 0.1 M KCl. Each symbol and color indicate a strain sweep as in Figure 4. Data are presented from a strain of 1% upward to keep visual clarity: (**a**) sample KIMN1; (**b**) sample KIMN2. All critical strains are those measured in a separate experiment and reported in Table 2.

**Figure 6 gels-10-00524-f006:**
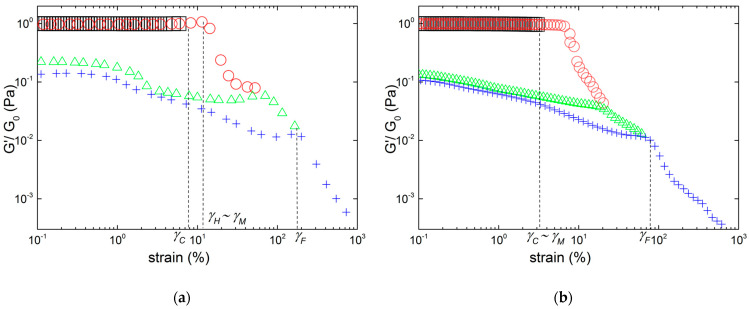
Successive LAOS sweeps performed on carrageen gels in 0.1 M KCl. Each symbol and color indicate a strain sweep as in Figure 4: (**a**) sample KIMN4; (**b**) sample K. All critical strains are those measured in a separate experiment and reported in Table 2.

**Figure 7 gels-10-00524-f007:**
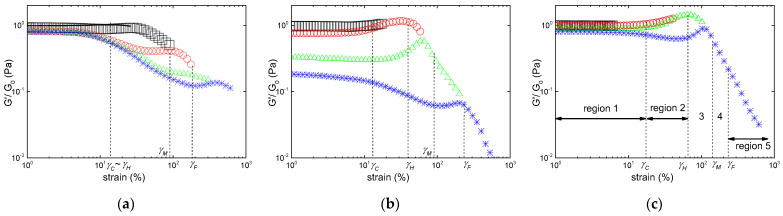
Successive LAOS sweeps performed on carrageen gels in 1 M NaCl. Each symbol and color indicate a strain sweep as in Figure 4: (**a**) sample KIMN1 with the line presenting the first strain sweep that stopped at *γ* > *γ_C_*; (**b**) sample KIMN4; (**c**) sample K+I. All critical strains are those measured in a separate experiment and reported in Table 3.

**Table 1 gels-10-00524-t001:** Characteristics of produced hybrid carrageenans and commercial kappa- and iota-carrageenans. Error bars are computed from duplicated analyses.

Samples	*ι* (mol.%)	*ν* and *μ* (mol.%)	*κ* (mol.%)	*M_w_* (×10^6^ g/mol)	*M_w_*/*M_n_*
KIMN1	78 ± 0.4	17.9 ± 2.1	4.1 ± 0.4	1.17	2.1
KIMN2	27.6 ± 1.3	23.3 ± 1.7	49.1 ± 0.7	0.85	2.1
KIMN3	33.3 ± 0.1	20.5 ± 0.1	46.1 ± 0.3	0.82	4.2
KIMN4	16.2 ± 0.7	11.8 ± 2.9	72.0 ± 0.8	1.27	2.9
I	92 ± 0.5	b.s. ^1^	8.0 ± 0.5	0.90	4.7
K	b.s. ^1^	b.s. ^1^	100 ± 0.5	1.13	2.7

^1^ b.s.—below the sensitivity of the ^1^H NMR spectrometer.

**Table 2 gels-10-00524-t002:** Rheological parameters computed from the analysis of the LAOS behavior for gels formed at 25 °C in 0.1 M KCl with 1 wt.% carrageenan. Errors are computed from duplicated experiments. The ratio of the storage modulus at the point of maximum *G′_max_* to the linear storage modulus *G*_0_ and the ratio of the loss modulus at the point of maximum *G″_max_* to the linear loss modulus *G″*_0_ are also reported to quantify the strength of the strain hardening.

Sample	*G*_0_ (Pa)	*γ_C_* (%) ^a^	*γ_H_* (%) ^a^	|*γ_C_* − *γ_H_*| (%)	*γ_M_* (%) ^a^	*γ_F_* (%) ^a^	*σ_F_* (Pa) ^a^	*R* (%)	*G′_max_/G* _0_	*G″_max_/G″* _0_
KIMN1	24.2 ± 0.6	39 ± 5	160 ± 10	121 ± 15	420 ± 30	480 ± 32	73.5 ± 0.5	90 ^c^	1.9	6.3
KIMN2	24 ± 11	7.0 ± 0.7	39 ± 5	32 ± 5.7	45 ± 6	108 ± 7	5.4 ± 0.3	53	2.2	3.4
KIMN3	24.9 ± 1.1	8.0 ± 2.0	29.5 ± 0.5	21.5 ± 2.5	45 ± 6	124 ± 8	3.6 ± 0.3	33	1.6	3.5
KIMN4	1164 ± 84	7.7 ± 1.1	11.6 ± 1.6	3.9 ± 2.7	12 ± 2	174± 22	9 ± 1	14	1.1	2.5
K	21,545 ± 1415	3.2 ± 2.2	n.a ^b^	n.a ^b^	3.0 ± 0.2	77 ± 10	116 ± 10	11	n.a ^b^	2.9
I	116.3 ± 4.6	39 ± 5	119 ± 3	89 ± 8	230 ± 30	245 ± 15	173 ± 10	100	1.6	13.5
K+I	7232 ± 297	10.1 ± 1.1	n.a ^b^	n.a ^b^	20 ± 3	520 ± 40	145 ± 10	7	n.a ^b^	1.8

^a^ Errors computed from duplicated tests or from the strain resolution in the logarithmic sweep with 20 strain values per decade. ^b^ n.a.—not available as corresponding gels are shear softening. ^c^ after sweeping up to a strain *γ_H_* < *γ* < *γ_M_*.

**Table 3 gels-10-00524-t003:** Rheological parameters computed from the analysis of the LAOS behavior for gels formed at 25 °C in 1 M NaCl with 1 wt.% carrageenan. Errors are computed from duplicated experiments. *γ_C_* is the strain signaling the end of the viscoelastic regime, *γ_H_* is the strain where *G′* shows a maximum in the strain hardening regime, *γ_M_* is the strain where *G″* shows a maximum (either during hardening or softening), and *γ_F_* is the strain where *G′* = *G″*, with the corresponding stress *σ_F_*. *R* is the elastic recovery of the gel defined by the ratio *G*_5_*/G*_0_, where *G_5_* is the linear storage modulus measured right after sweeping the strain up to *γ_F_*. The ratio of the storage modulus at the point of maximum *G′_max_* to the linear storage modulus *G*_0_ and the ratio of the loss modulus at the point of maximum *G″_max_* to the linear loss modulus *G″*_0_ are also reported to quantify the strength of the strain hardening.

Sample	*G*_0_ (Pa)	*γ_C_* (%) ^a^	*γ_H_* (%) ^a^	|*γ_C_* − *γ_H_*|(%)	*γ_M_* (%) ^a^	*γ_F_* (%) ^a^	*σ_F_* (Pa) ^a^	*R* (%)	*G′_max_/G* _0_	*G″_max_/G″* _0_
KIMN1	41 ± 3	14 ± 1	14 ± 1	0 ± 2	95 ± 10	185 ± 15	21 ± 2	80	1.04	1.8
KIMN4	285 ± 5	12.5 ± 2.5	40 ± 5	27.5 ± 7.5	88 ± 12	230 ± 30	127 ± 10	18	1.6	2.4
K	665 ± 15	1.6 ± 0.6	n.a. ^b^	n.a. ^b^	7.7 ± 1.1	>512	>59	17	n.a. ^b^	1.1
I	95 ± 10	61 ± 30	85 ± 5	24 ± 35	15 ± 7	245 ± 15	55 ± 2	n.a. ^b^	1.05	6
K+I	159 ± 1	17.8 ± 1.6	64.5 ± 7.5	82.3 ± 9.1	145 ± 10	235 ± 30	48 ± 10	81 ^c^	1.4	2.1

^a^ Errors computed from duplicated tests or from the strain resolution in the logarithmic sweep with 20 strain values per decade. ^b^ n.a.—not available. ^c^ after sweeping up to a strain *γ_H_* < *γ* < *γ_M_*.

## Data Availability

The raw data supporting the conclusions of this article will be made available by the authors on request.

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
