# Peer review of "Viscoelastic Reversibility of Carrageenan Hydrogels under Large Amplitude Oscillatory Shear: Hybrid Carrageenans versus Blends"

_gels, 2024, doi:10.3390/gels10080524_

Round 1

Reviewer 1 Report

Comments and Suggestions for Authors

Although carrageenan is widely used in food industry, the viscoelastic response of carrageenan hydrogels to large amplitude oscillatory shear (LAOS) has not received attention in the literature. The authors make great contribution to the work that a set of hybrid-carrageenans with differing chemical compositions are gelled in the presence of KCl or NaCl, and their nonlinear viscoelastic responses are systematically compared with mixtures of kappa- and iota-carrageenans of equivalent kappa-carrageenan contents. This study is very interesting due to modification of the carrageenan hydrogels, and a great deal of experimental work was done to verify the above ideas.

1)     How to prepare for the samples of KIMN1, KIMN2, KIMN3, and KIMN4? In the lines 91-123, or the materials and methods section, there was no introduction for these four samples.

2)     Line 81-88, the expected aim of the present study should be clear given. The hybrid carrageenans were formed in the presence of 1 M NaCl or 0.1 M KCl, the reason for use of KCl and the concentration range should be given.

3)     In tables 1-3, the multiple comparisons should be given,

4)     In table 2, the range of rheological parameters had better be given in Materials and methods section or the text, thus more readers can compare their results with the present results.

5)     Line 94, the title of X-axis should be given.

Author Response

COMMENT1)     How to prepare for the samples of KIMN1, KIMN2, KIMN3, and KIMN4? In the lines 91-123, or the materials and methods section, there was no introduction for these four samples.

Answer 1): We add the following lines right at the beginning of the section to explain how samples were prepared, though the full experimental detail is given in section 4: “Hybrid-carrageenans were hot extracted from commercial seaweeds in deionized water, see full experimental detail in section 4.2. For NMR measurements, solutions of extracts in D2O were prepared. To limit line broadening in NMR spectra, these solutions were ultrasonicated during few hours before filling NMR tubes and acquiring NMR spectra at 70 ºC. Full experimental detail is given in section 4.2.”.

COMMENT 2)     Line 81-88, the expected aim of the present study should be clear given. The hybrid carrageenans were formed in the presence of 1 M NaCl or 0.1 M KCl, the reason for use of KCl and the concentration range should be given.

Answer 2): Lines 81-88 have been reformulated to clarify the choice for cation, ionic strength and carrageenan concentration, as well as the expected aim of the study. These lines now read: “Because the LAOS behavior of various carrageenan gels is the focus of the present study, the type of salt and ionic strength, needed to form a gel with large enough elasticity for rotational rheometry for all types of studied carrageenans, are to be established. Earlier studies on the phase diagrams of KIMN and K+I blends showed that gels could be formed with a carrageenan concentration of 1 wt.% and 1 M NaCl or 0.1 M KCL . Thus, such gelling conditions were used to report on the LAOS response and the LAOS reversibility of the 4 types of carrageenan gels.”.

COMMENT 3)     In tables 1-3, the multiple comparisons should be given,

Answer 3): we understand that this reviewer is questioning the origin of the errors’ computation. We provided such information by adding the following line in the title of the table1: “Error bars are computed from duplicated analyses.”.  Note that for tables 2 and 3, such information is given by the footnote a, which reads: “Errors computed from duplicated tests or from the strain resolution in the logarithmic sweep with 20 strain values per decade.”.

COMMENT 4)     In table 2, the range of rheological parameters had better be given in Materials and methods section or the text, thus more readers can compare their results with the present results.

Answer 4): we shifted the range of rheological parameters (strains) in the Materials and method section. Note that in the text, the discussion of Figure 2 gives the full range of rheological parameters, which facilitates the reading of the information contained in Table 2.

COMMENT 5)     Line 94, the title of X-axis should be given.

Answer 5): the title “chemical shift (ppm)” is now given in the new Figure 1.

Reviewer 2 Report

Comments and Suggestions for Authors

The article by Moraes I.C.F. and Hilliou L. describes the nonlinear viscoelasticity for hydrogels of carrageenans differing in chemical structure. The authors use four samples of carrageenan extracted by them from algae, two commercial samples, and their equimass blend, producing hydrogels of equal concentration in aqueous solutions of sodium chloride or potassium chloride for comparative tests on shear softening, shear hardening, and the ability to recover structure after shear. As a result, the authors make conclusions about the relationships between the rheological behavior of carrageenan hydrogels and their macromolecular structure. The article is well-written and contains high-quality experimental work, which can be published after some improvements.

Specific comments are as follows.

Lines 38, 415: “a statistical block copolymer”, “in the block copolymer”. Copolymers can be block copolymers, multi-block copolymers, and statistical copolymers. A statistical block copolymer sounds like nonsense. What do the authors want to say? In addition, are the authors sure when writing about the block structure of carrageenan? Maybe the authors mix up blocks and disaccharide units.

Line 39 and below: “alternating 3-linked b-D-galactopyranose (G-units) and 4-39 linked a-D-galactopyranose (D-units) or 4-linked 3,6-anhydrogalactose”. The difference in the chemical structure of saccharides and carrageenans based on them can be illustrated in Figure 1b for clarity.

Line 56: “their nonlinear rheological behavior has not received much attention 56 in the literature”. The nonlinear rheological behavior of kappa-carrageenan gel in a blend with gelatin was also studied in 10.1016/j.lwt.2015.03.024.

Line 83: “KCL” -> “KCl”.

Lines 171, 184: “The strain hardening illustrated in Figure 2”, “The LAOS behavior of the kappa-carrageenan gel portrayed in Figure 3 is different from the ones displayed in Figure 2.” Previously, it was proposed to call such behaviors as strong strain overshoot and weak strain overshoot, respectively (rather than strain hardening), see, e.g., the review in 10.1016/j.progpolymsci.2011.02.002.

Line 296: “sample KIMN4 shows a much smaller strain hardening than the one”. It would have been helpful if the authors had expressed the strength of strain hardening quantitatively, e.g., by adding the ratio of the storage modulus at the point of maximum to the linear storage modulus (G'max/G'0) in Tables 2 and 3. The same can be done for the loss modulus G"max/G"0.

Author Response

The article by Moraes I.C.F. and Hilliou L. describes the nonlinear viscoelasticity for hydrogels of carrageenans differing in chemical structure. The authors use four samples of carrageenan extracted by them from algae, two commercial samples, and their equimass blend, producing hydrogels of equal concentration in aqueous solutions of sodium chloride or potassium chloride for comparative tests on shear softening, shear hardening, and the ability to recover structure after shear. As a result, the authors make conclusions about the relationships between the rheological behavior of carrageenan hydrogels and their macromolecular structure. The article is well-written and contains high-quality experimental work, which can be published after some improvements.

Specific comments are as follows.

COMMENT 1: Lines 38, 415: “a statistical block copolymer”, “in the block copolymer”. Copolymers can be block copolymers, multi-block copolymers, and statistical copolymers. A statistical block copolymer sounds like nonsense. What do the authors want to say? In addition, are the authors sure when writing about the block structure of carrageenan? Maybe the authors mix up blocks and disaccharide units.

ANSWER 1: we clarified the copolymer structure of carrageenans by rewording line 38 as follows: “These biopolymers show a complex chemical structure, which is better described as a multi-block copolymer made of statistically distributed sequences (blocks) of alternating”. Indeed, one block is made of several disaccharide units of a carrageenan type. The position and length of the block in the polymer chain follows a statistical distribution.

COMMENT 2: Line 39 and below: “alternating 3-linked b-D-galactopyranose (G-units) and 4-39 linked a-D-galactopyranose (D-units) or 4-linked 3,6-anhydrogalactose”. The difference in the chemical structure of saccharides and carrageenans based on them can be illustrated in Figure 1b for clarity.

ANSWER 2: a new Figure 1a which displays the chemical structure is now added, before Figure 1 which is now 1b.

COMMENT 3: Line 56: “their nonlinear rheological behavior has not received much attention 56 in the literature”. The nonlinear rheological behavior of kappa-carrageenan gel in a blend with gelatin was also studied in 10.1016/j.lwt.2015.03.024.

ANSWER 3: We thank the reviewer for bringing this study to our knowledge. As we feel that the topic of this referenced paper is out of the scope of our present study, which focuses on carrageenan, not on gelatin and its specific issues, and because reviewer 1 asked for clarity in the presentation of the object of this study, we choose to not consider this suggestion in the introduction.

COMMENT 4: Line 83: “KCL” -> “KCl”.

ANSWER 4: thank you for the alert. This is now corrected.

COMMENT 5: Lines 171, 184: “The strain hardening illustrated in Figure 2”, “The LAOS behavior of the kappa-carrageenan gel portrayed in Figure 3 is different from the ones displayed in Figure 2.” Previously, it was proposed to call such behaviors as strong strain overshoot and weak strain overshoot, respectively (rather than strain hardening), see, e.g., the review in 10.1016/j.progpolymsci.2011.02.002.

ANSWER 5: we added the following 2 lines to refer to the labeling of the suggested review which is now included in the cited literature of the paper: “Note here that the strain hardening depicted in Figure 2 was labelled elsewhere as a strong strain overshoot [27], to qualify the LAOS response of various materials without any relation with the underlying hardening structure.” and “Such behavior was labelled as weak strain overshoot elsewhere [27].”.

COMMENT 6: Line 296: “sample KIMN4 shows a much smaller strain hardening than the one”. It would have been helpful if the authors had expressed the strength of strain hardening quantitatively, e.g., by adding the ratio of the storage modulus at the point of maximum to the linear storage modulus (G'max/G'0) in Tables 2 and 3. The same can be done for the loss modulus G"max/G"0.

ANSWER 6: we appreciate the suggestion. Two columns are now added to the tables with the following line in the titles to tables: “The ratio of the storage modulus at the point of maximum G’max to the linear storage modulus G0 and the ratio of the loss modulus at the point of maximum G’’max to the linear loss modulus G’’0 are also reported to quantify the strength of the strain hardening.”..